# Travel Characteristics Analysis and Passenger Flow Prediction of Intercity Shuttles in the Pearl River Delta on Holidays

**Binglei Xie [1], Yu Sun [1], Xiaolong Huang [1], Le Yu [1,2] and Gangyan Xu [1,*]**

[1] School of Architecture, Harbin Institute of Technology (Shenzhen), Shenzhen 518000, China; xbl@hit.edu.cn (B.X.); 18b932019@stu.hit.edu.cn (Y.S.); huangxiaolong@stu.hit.edu.cn (X.H.); yule@stu.hit.edu.cn (L.Y.)

[2] Department of Building and Real Estate and Research Institute of Sustainable Development, The Hong Kong Polytechnic University, Hong Kong 999077, China

\* Correspondence: gangyan@hit.edu.cn; Tel.: +86-755-26033792

**Abstract:** As China's urbanization process continues to accelerate, the demand for intercity residents' transportation has increased dramatically. Holiday travel has different demand characteristics, causing serious shortage during peak periods. However, current research barely focuses on the passenger flow prediction along with travel characteristics of intercity shuttles. Accurately predicting passenger flow during the holidays helps to improve operational organization efficiency and residents' satisfaction, and provides a basis for reasonable resource allocation by the management department. This paper analyzes the spatiotemporal characteristics of intercity shuttles passenger flow in the Pearl River Delta. Separate passenger flow prediction models on non-holiday and holiday are established using an improved genetic algorithm optimized back propagation neural network (IGA-BPNN) based on the characteristics of passenger flow, and the prediction models are validated based on panel data. The results of weekly flow show obvious holiday characteristics, and the hourly traffic flow of holidays is much larger than that of weekends and weekdays. There is a significant difference in the hourly flow between different holidays. The IGA-BPNN model used in this paper achieves lower prediction error relative to the benchmark BPNN approach (leads a two thirds reduction in MAPE, and an over 85% reduction in MSPE).

**Keywords:** holidays; intercity shuttles; travel characteristics; passenger flow prediction; improved BP neural network

## 1. Introduction

China's urbanization process has been accelerating, and the boundaries between cities have blurred increasingly. Urban clusters such as the Pearl River Delta have gradually formed. It is the core area of Guangdong Province in the Guangdong-Hong Kong-Macao Bay Area. The road network structure is relatively developed, and the density of expressways ranks first in Asia. Nowadays, the traffic flow of multiple expressways in the Pearl River Delta is increasing, and some transportation hubs such as Guangzhou South station are already saturated [1], and problems such as traffic congestion and decline in service capacity continue to emerge. It is urgent to solve the problems of regional transportation organization. Close intercity connections generate a large amount of traffic demand especially during legal holidays. The intercity shuttle is one of the main intercity transportation modes. Due to the large travel demand of intercity travel and the uneven travel distribution on time and space, some hotspot intercity shuttle tickets are hard to get during holidays. The phenomenon of the excess capacity of intercity shuttles during normal periods and severely insufficient capacity

during peak periods, seriously reduces the service quality of transportation companies and passenger travel satisfaction.

In order to improve the efficiency of intercity passenger transportation, it is necessary to master the space-time variation and then predict the passenger flow. However, current research seldom concentrates on passenger flow prediction about intercity shuttles. It may be due to the difficulty of collecting intercity travel data or the limitation of prediction models. Operating companies can reserve the required vehicles and personnel in advance according to the specific passenger flow growth. They can transfer vehicles from lines with lower passenger demand to higher ones, and adjust the existing passenger departure schedules to respond to peak passenger flows during holidays more efficiently and accurately.

This paper takes the passenger flow data of the intercity shuttle as the research object based on a megalopolis like the Pearl River Delta and studies the space-time characteristics of the intercity shuttle passenger flow. Improved back propagation (BP) neural network is used to predict the passenger flow, which is essential for improving the organization efficiency of intercity shuttles in megalopolises, and also offers a reliable basis for the operation and scheduling of intercity shuttles, providing an effective reference for traffic planning and transportation organizations between urban groups.

The daily passenger flow of the intercity shuttle fluctuates violently and has typical non-linearity and non-stationarity characteristics. The traditional linear passenger flow forecasting model such as a simple autoregressive or ARIMA model has poor prediction accuracy when data fluctuates, thus it is not applicable. BP neural network is a supervised learning algorithm with a strong nonlinear prediction ability and a flexible network structure using error back propagation algorithm, and it is the most widely used neural network at present with multiple successful applications. Therefore, this paper uses BP neural network as the basis for passenger flow forecasting of intercity passenger train lines.

This paper divides the passenger flow prediction into non-holiday and holiday passenger flow prediction, and proposes non-holiday passenger flow prediction model coupling with a genetic algorithm optimized BP neural network based on the spatial and temporal characteristics of passenger flow. Then, after considering the characteristics of holiday passenger flow, a holiday passenger flow prediction method based on the holiday background passenger flow and holiday fluctuation coefficient is proposed, realizing relatively accurate predictions of holiday passenger flow when there is little historical data.

Theoretically, the existing passenger flow prediction research mostly deals with railway passenger flow, conventional bus passenger flow, and urban rail transit passenger flow [2–20]. This paper takes the passenger flow data of the intercity shuttle as the research object. Moreover, the research range is based on the urban agglomeration, and studies the intercity shuttle passenger flow on holidays. This study can contribute to make up for the gap in passenger flow forecasting of intercity shuttles, to improve relevant research on holiday passenger flow prediction and enrich the theoretical system of passenger flow prediction to a certain degree.

Practically, this article accurately predicts passenger flow of intercity shuttles and grasps non-holiday and holiday travel demand. The research can provide a reference for the operation and scheduling of passenger transport enterprises and a basis for the optimal management of passenger stations which not only improves transportation efficiency but also saves the operating costs to a certain extent. It can also provide a reference for travelers to choose the travel mode. This has extremely important practical significance for alleviating the intercity travel difficulties and improving the intercity travel happiness of passenger urban agglomerations.

The goal of the paper can be decomposed into three aspects: First, analyze the characteristics of time and space changes in passenger flow of intercity shuttles, and clarify the different characteristics between holidays and non-holiday intercity travel. Second, according to the factors affecting the intercity shuttles travel, a non-holiday passenger flow forecasting model system based on the BP neural network coupling with improved genetic algorithm is established (improved genetic algorithm-BP neural network—IGA-BPNN). Third, examine the influence mechanism of holidays on intercity travel, establish a passenger flow forecasting model considering the influence of holidays, which provides a theoretical basis and methodological basis for the allocation of transportation resources.

The organization of the paper is as follows. Section 2 reviews existing research of short-term passenger flow prediction on non-holiday and holidays. Section 3 extracts spatio-temporal features of intercity shuttles passenger flows in the Pearl River Delta in China. Section 4 proposes the IGA-BPNN model to predict the non-holiday passenger flow of the Shenzhen–Guangzhou line and analyzes its prediction result. Section 5 implements the holiday passenger flow prediction model based on the former approach and explains its application. Section 6 concludes this paper.

## 2. Literature Review

### 2.1. Passenger Flow Prediction Research

The following reviews the relevant research on passenger flow prediction from three aspects: Research on railway passenger flow, conventional bus passenger flow and urban rail transit passenger flow.

In terms of railway passenger flow prediction, Tsai et al. studied the neural network models of two short-term railway passenger flow predictions. The prediction accuracy of the two neural network structures is better than that of the traditional multi-layer perceptron [2]; Alexander et al. reviewed and analyzed the shortcomings and potential problems of the high-speed railway passenger flow regression model, proposed the passenger flow prediction method considering the economic development level along the railway [3]; Dou et al. proposed a high-speed railway passenger flow approach based on fuzzy time logic, significantly improved the accuracy of the prediction model [4]. Xu et al. proposed a method based on spatio-temporal data mining to predict railway passenger flow by analyzing the limitations of existing prediction methods and used this method to predict railway passenger flow during the Spring Festival. The prediction accuracy is ideal [5]. Jiang et al. analyzed the holiday passenger flow characteristics of the three-day holiday and the seven-day holiday. A hybrid short-term daily passenger flow prediction method is proposed combining the ensemble empirical mode decomposition (EEMD) and the grey support vector machine (GSVM) model. The model had good prediction accuracy as well, and it was especially suitable for short-term high-speed passenger flow prediction [6]; Huang et al. proposed the neural network short-term railway passenger flow prediction method based on radial basis function (RBF), and using it in the holiday passenger flow prediction experiment, proved that the model is faster than the BP neural network, and the prediction accuracy was relatively high [7].

In terms of conventional bus passenger flow prediction, Bai et al. proposed a multi-mode deep fusion short-term prediction algorithm. The results showed that the model's average absolute error is better than others [8]; Zhang et al. proposed a feature-based analysis method of Karlman filter to predict short-time bus passenger flow, and the results are more accurate than the artificial neural network [9]; Ma et al. proposed a hybrid prediction method based on interactive multi-model to achieve the maximum model performance [10]. Lu et al. introduced the concept of pan-holiday, taking the preferential promotion day as a new festival to analyze the spatial and temporal distribution characteristics of bus passenger flow in Huangshan Scenic Area [11]; Liu et al. proposed a public network based on deep neural network. The model used the passenger flow data of Taipei City to input training on various combinations of historical passenger flow, time, direction, and holidays. The experimental results showed that the modeling method and input influencing factors are all important factors affecting the performance of the passenger flow prediction model [12]; Zhang et al. analyzed the impact of weather, environment, holidays, and other factors on bus passenger flow in continuous large-scale activities. The analysis showed that tourism cost, weather, and holidays were the main factors affecting passenger traffic volume, and the establishment of the model laid a solid theoretical foundation for the establishment of the model [13].

In the aspect of urban rail transit passenger flow prediction, Roos et al. proposed a rail transit passenger flow prediction algorithm based on dynamic Bayesian network and Gaussian mixture model, which is applied to passenger flow prediction of Paris Metro Line 2 [14]; Li et al. proposed a hybrid prediction model combining symbolic regression and autoregressive comprehensive moving average

model to improve the prediction accuracy and interpretability of traditional models [15]; Yu et al. proposed an EMD-BPNN prediction method combining empirical mode decomposition (EMD) and BP neural network (BPNN) to predict the short-term passenger flow of the subway system [16]. Yang and Wu considered the influence of weather, date, and other factors, analyzed the prediction accuracy of different training time and learning speed to determine the optimal number of neurons in the hidden layer, and proposed a BP neural network prediction method [17]. Li analyzed the data characteristics of the passenger flow of the National Day holiday of Guangzhou Metro Line 5, and proposed the passenger flow prediction method combining the time series model and the regression model [18]; Zhu introduced the urban rail transit daily passenger flow based on the average daily passenger flow index, the ARIMA model of urban rail transit daily passenger flow prediction was established, and the passenger flow before and after holidays and during holidays were predicted. The results showed that the relative error of predicted passenger flow was about 2% [19]. Ma et al. conducted a sensitivity analysis of urban rail transit passenger flow, focusing on the impact of holiday passenger flow fluctuation on prediction accuracy, and proposed a solution to smooth decomposition of holiday wave passenger flow [20].

Ghalehkhondabi reviewed the demand forecasting methods within tourism passenger transportation [21], and the main methods include time series models [22–25], autoregressive moving average (ARMA), autoregressive integrated moving average (ARIMA), and the seasonal ARIMA(SARIMA) [26–28], regression models [29–31], support vector machines [32–34], artificial neural networks (ANN) models [23,35,36].

Research of Mario proved the impact of public holiday [37], proposing that the daily traffic counts had obvious weekly variation cycles by using time series approach [38] or ARIMAX and SARIMAX models [39].

According the study of Chen the application of back-propagation neural networks improved the forecasting accuracy of air passenger demand with mean absolute percentage error (MAPE) of 0.34% [40]. Blinova developed the neural network method using 28 time-lagged feed-forward artificial neural networks to forecast air passenger traffic flows in Russia [41]. Other scholars also use neural network methods to predict short-term air transportation demand [42] or predict bus arrival times [43], sometimes in other prediction problem [44].

The peak load of a bus route is essential to service frequency setting [45]. Transit network planning requires prediction of state variables such as on-board loads [46]. Generally, the methods to predict bus on-board loads can be classified into two types—model-based approach [47] and simulation-based approach [48]. Other researchers also predict the real-time congestion information of the subway [49].

## 2.2. Summary of the Research

Firstly, the existing research about passenger flow prediction mostly deals with railway, conventional bus, and urban rail transit passenger flows, and there is less research on passenger flow prediction of intercity shuttle. It is more difficult to obtain history data of intercity shuttle than that of railways, conventional buses, and urban rail transit due to the more complicated operation subject it involves. Therefore, the relevant theoretical system has not yet been formed.

Secondly, the prediction model of current research has lower prediction accuracy and poor model applicability. Some scholars use a single model such as BP neural network and the k nearest Neighbor (KNN) to predict passenger flow but do not consider the large fluctuation of passenger flow and the defects of the single prediction model; relevant scholars perform linear or nonlinear fitting only based on historical passenger flow data, completely ignoring the relevant spatiotemporal features of passenger flow. Some scholars slightly consider the spatial and temporal characteristics of passenger flow, but still have not grasped the key features.

Moreover, most studies lack the consideration of holiday passenger flow characteristics, and have not established a passenger flow prediction model specifically for holidays. The predicted results are significantly different from the actual passenger flow, and the model's prediction effect is not good.

Some scholars directly apply the method of non-holiday passenger flow prediction to predict holiday passenger flow, therefore, the results cannot provide a reliable reference for decision-making.

This paper analyzes the passenger flow characteristics of the holiday, predicted the non-holiday and holiday passenger flow of the intercity shuttle using the BP neural network coupling with improved genetic algorithm.

The BP neural network needs to randomly assign initial connection weights and thresholds for each layer when starting training, and the BP neural network is more sensitive to the selection of the initial weights and thresholds. When they are not selected properly, it will lead to slower convergence speed and local extreme values, which will affect the calculation efficiency and prediction accuracy of BP neural network, and it may cause its prediction accuracy to be lower than that of traditional linear prediction models in severe cases. Some scholars have proposed to optimize the initial weights and thresholds of BP neural networks based on genetic algorithms using traditional roulette and tournament methods in the selection operation. The method of operation does not improve the fitness function, with the disadvantages of slow convergence and easiness to fall into the local optimum when the traditional genetic algorithm is optimized. Therefore, its prediction effect is still not ideal, and the prediction accuracy needs to be improved.

According to the passenger flow characteristics of the intercity shuttle, the passenger flow is predicted and studied, and the obtained non-holiday and holiday passenger flows of the intercity shuttle can provide corresponding reference for the intercity traveler. It provides an important theoretical and practical significance for providing reliable passenger flow basis for the operation and scheduling of intercity passenger transport lines in a megalopolis.

## 3. Passenger Flow Characteristic

### 3.1. Data Introduction and Preprocessing

The data in this paper is the outbound passenger flow data of each passenger line in Guangdong Province provided by the Guangdong Provincial Department of Transportation and the passenger shift data obtained on the online ticketing official website of Guangdong Province. The outbound passenger flow data includes the outbound passenger flow data of each passenger line in Guangdong Province in 2017–2018, totaling more than 8400 passenger lines and more than 14 million records, including seven fields such as the day of the week, time of day(TOD), outbound time, line name, starting point, ending point, the total number of seats, and total number of outbound stations; the passenger shift data on the online ticketing official website of Guangdong Province includes 12 fields as the shift number, departure station, destination, departure time, passenger type, seat type, vehicle level, arrival station, route station, mileage, fare, and ticket sales. The two kinds of data are stored in the SQL (Structured Quevy Language) Server 2012 database to clean, query, and analyze. The null value data and the invalid data are replaced by the average value of the passenger bus off-site passenger flow data. The outbound passenger flow data format and example are shown in Table 1.

**Table 1.** Outbound passenger flow data format and example.

| The Day of the Week | TOD (Time of Day) | Line Name | Starting Station | Terminal | Seats | Passenger Number |
|---|---|---|---|---|---|---|
| Thursday | 08:25:00 | Yingde-Guangzhou | Qingyuan Yingde Bus Terminal | Panyu Bus Terminal | 47 | 7 |
| Sunday | 08:25:00 | Yingde-Guangzhou | Qingyuan Yingde Bus Terminal | Panyu Bus Terminal | 47 | 47 |
| Wednesday | 08:25:00 | Yingde-Guangzhou | Qingyuan Yingde Bus Terminal | Panyu Bus Terminal | 47 | 19 |

### 3.2. Space Characteristics

Spatial characteristics include three parts: Spatial shift characteristics, spatial flow characteristics, spatial passenger flow characteristics in holidays.

### 3.2.1. Spatial Shift Characteristics

The average daily departures of intercity shuttle in the Pearl River Delta in 2017–2018 is counted. The data in Table 2 shows that in the Pearl River Delta region, Guangzhou, Shenzhen, Foshan, Dongguan, and other cities with strong economic strength have more intercity shuttle buses, while the frequency of other cities is relatively low, indicating a serious imbalance.

**Table 2.** Spatial shift characteristics (departures).

|  | Guangzhou | Shenzhen | Zhuhai | Foshan | Huizhou | Dongguan | Zhongshan | Jiangmen | Zhaoqing |
|---|---|---|---|---|---|---|---|---|---|
| Guangzhou | - | 451 | 239 | 473 | 265 | 411 | 241 | 264 | 149 |
| Shenzhen | 437 | - | 196 | 239 | 331 | 365 | 181 | 114 | 96 |
| Zhuhai | 248 | 188 | - | 160 | 89 | 105 | 142 | 153 | 38 |
| Foshan | 485 | 233 | 152 | - | 131 | 192 | 138 | 144 | 254 |
| Huizhou | 255 | 357 | 87 | 127 | - | 150 | 70 | 48 | 75 |
| Dongguan | 427 | 357 | 103 | 195 | 157 | - | 103 | 63 | 59 |
| Zhongshan | 223 | 125 | 146 | 133 | 75 | 100 | - | 106 | 51 |
| Jiangmen | 267 | 119 | 150 | 143 | 46 | 61 | 111 | - | 63 |
| Zhaoqing | 157 | 91 | 36 | 241 | 78 | 61 | 50 | 65 | - |

The origin and destination of the intercity shuttle bus cannot be in the same city, so there is "-" between the same cities.

Take the hourly passenger flow of a typical line (Guangzhou–Shenzhen and Shenzhen–Guangzhou) as an example to study the difference between the opposite directions. The 15 time periods departures in the operating hours from 6:00 to 21:00 are counted. The Guangzhou–Shenzhen intercity shuttle has a total of 451 departures, while Shenzhen–Guangzhou has 437, slightly less than the Shenzhen–Guangzhou intercity line.

### 3.2.2. Spatial Flow Characteristics

According to the departure passenger flow data and the starting and terminal station of the line, the annual passenger flow the origin and destination (OD) data of the intercity shuttle in the Pearl River Delta in 2018 is shown in Table 3.

**Table 3.** Annual origin and destination (OD) passenger flow of intercity shuttle line among nine cities in the Pearl River Delta (10,000 persons).

|  | Guangzhou | Shenzhen | Zhuhai | Foshan | Huizhou | Dongguan | Zhongshan | Jiangmen | Zhaoqing |
|---|---|---|---|---|---|---|---|---|---|
| Guangzhou | - | 181.54 | 85.44 | 190.64 | 113.31 | 145.01 | 83.05 | 99.86 | 45.30 |
| Shenzhen | 178.68 | - | 62.30 | 100.23 | 165.24 | 184.24 | 59.29 | 41.88 | 34.50 |
| Zhuhai | 83.08 | 61.59 | - | 50.57 | 30.31 | 34.50 | 54.21 | 65.25 | 11.25 |
| Foshan | 203.55 | 110.25 | 48.26 | - | 40.55 | 67.58 | 53.27 | 55.31 | 98.53 |
| Huizhou | 117.18 | 162.35 | 29.89 | 41.84 | - | 52.40 | 24.90 | 19.29 | 22.36 |
| Dongguan | 163.24 | 179.57 | 31.23 | 66.25 | 50.79 | - | 40.57 | 28.74 | 25.34 |
| Zhongshan | 80.39 | 58.12 | 55.33 | 54.79 | 23.99 | 39.34 | - | 42.20 | 20.24 |
| Jiangmen | 105.79 | 43.56 | 63.14 | 53.29 | 20.23 | 27.70 | 44.29 | - | 26.79 |
| Zhaoqing | 47.05 | 33.30 | 12.46 | 99.34 | 21.58 | 26.79 | 21.54 | 27.48 | - |

The origin and destination of the intercity shuttle bus cannot be in the same city, so there is "-" between the same cities.

It can be seen from Table 3 that the passenger flow has significant spatial heterogeneity. Passenger flow is mainly on 10 important lines such as Guangzhou and Shenzhen, Guangzhou and Foshan, and Guangzhou and Dongguan, Shenzhen and Dongguan, Shenzhen and Huizhou. There is less traffic in other directions, and the annual passenger flow between different intercity shuttle lines are quite different.

### 3.2.3. Spatial Passenger Flow Characteristics in Holidays

Holidays in this article refer to the seven statutory holidays prescribed by the State Council: New Year's Day, Spring Festival, Ching Ming Festival, Labor Day, Dragon Boat Festival, Mid-Autumn Festival, and National Day. Special circumstances such as holiday extension and holiday overlap are not considered in this article. Calculate the passenger flow of intercity shuttle among nine cities in

the Pearl River Delta during the holidays in 2018. The passenger flow of each intercity shuttle in the Spring Festival holiday (15 February to 21 February) is shown in Table 4.

**Table 4.** Passenger flow of each intercity shuttle in the Spring Festival (person).

|  | Guangzhou | Shenzhen | Zhuhai | Foshan | Huizhou | Dongguan | Zhongshan | Jiangmen | Zhaoqing |
|---|---|---|---|---|---|---|---|---|---|
| Guangzhou | - | 32,986 | 16,384 | 30,587 | 19,827 | 22,023 | 15,367 | 20,225 | 11,234 |
| Shenzhen | 32,338 | - | 10,235 | 21,005 | 24,253 | 23,598 | 11,023 | 13,026 | 10,324 |
| Zhuhai | 15,268 | 13,524 | - | 8805 | 6637 | 6549 | 9724 | 9996 | 3031 |
| Foshan | 31,598 | 20,369 | 8726 | - | 5049 | 5538 | 5234 | 5367 | 10,230 |
| Huizhou | 20,285 | 23,005 | 6029 | 5837 | - | 7785 | 4027 | 3939 | 4127 |
| Dongguan | 23,056 | 22,583 | 6387 | 5347 | 7584 | - | 6631 | 5031 | 4704 |
| Zhongshan | 16,304 | 10,354 | 9596 | 5108 | 4178 | 6541 | - | 5029 | 2751 |
| Jiangmen | 23,589 | 14,037 | 10,123 | 5444 | 4052 | 4501 | 5523 | - | 4021 |
| Zhaoqing | 10,258 | 9992 | 2968 | 11,093 | 3987 | 4555 | 2689 | 3825 | - |

The origin and destination of the intercity shuttle bus cannot be in the same city, so there is "-" between the same cities.

The passenger flows on other holidays were similar to that of the Spring Festival. According to the passenger flow in Table 4, there are significant differences in the distribution characteristics of passenger flow in the intercity shuttle between different holidays. Passenger flow on New Year's Day, Ching Ming Festival, Labor Day, Dragon Boat Festival, and Mid-Autumn Festival holiday are not much different, the urban intercity travel demand of them are generally large; passenger flow on National Day holiday is huge, the passengers demand for intercity travel has increased dramatically.

### 3.3. Time Characteristics

Due to the different travel needs of travelers in different time periods, the same passenger transport line will present different passenger flow characteristics in different time scales. This section will study the passenger flow data of Shenzhen–Guangzhou intercity shuttle in 2017–2018. The passenger flow characteristics of the intercity shuttle on different time scales (quarter, month, week, day, and hour) are explored.

### 3.3.1. Daily Variation Characteristics

The daily traffic of 2018 is classified according to the "day of the week", with 52 Tuesdays to Sundays and 53 Mondays. The maximum, minimum, average, and variance of the passenger traffic for each day from Monday to Sunday are counted separately.

The results are shown in Table 5.

**Table 5.** Statistics of passenger flow during the week of 2018 Shenzhen–Guangzhou.

|  | Minimum Flow (Person) | Maximum Flow (Person) | Average Flow (Person) | Standard Deviation |
|---|---|---|---|---|
| Monday | 3337 | 11,438 | 4856 | 1618 |
| Tuesday | 2638 | 9381 | 4253 | 1407 |
| Wednesday | 2864 | 8398 | 4025 | 1168 |
| Thursday | 2742 | 8822 | 3993 | 1075 |
| Friday | 2583 | 8470 | 4914 | 951 |
| Saturday | 3390 | 11,316 | 5523 | 1256 |
| Sunday | 3240 | 9550 | 5797 | 1084 |

At the same time, in order to explore the passenger flow change laws during the week, the passenger flow in July 2018 is divided into four weeks, and the variation of passenger flow in the Shenzhen–Guangzhou intercity shuttle line is explored. Figure 1 presents a typical "three-stage": Stage 1, a downward trend from Monday to Tuesday; Stage 2, a relatively stable state from Tuesday to Thursday; Stage 3, a sharp upward trend from Thursday to Sunday. Different from the change of passenger flow in

the city (large passenger flow on weekdays and small flow during the weekend), the intercity shuttle is just the opposite (the passenger flow is greater on the weekend than the weekdays).

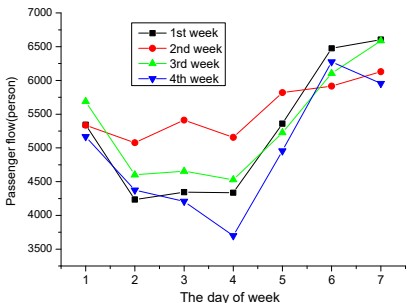

**Figure 1.** Daily passenger flow (weekly).

As shown in Table 5, there are large differences in passenger flow between different "day of the week". Monday's average passenger traffic ranked fourth, but its passenger traffic fluctuated the most. The passenger flow characteristics on Tuesday, Wednesday, and Thursday are similar, and the average value is small. The daily traffic flow is also less volatile; Friday is more special, the average daily passenger flow is in the third place, but its passenger flow is the least volatile; the average passenger flow on Saturday is less than Sunday, but the daily passenger flow on Sunday fluctuates more than Saturday. In summary, the "day of the week" feature of daily passenger flow is more obvious, and it is necessary to consider the "day of the week" factor in the prediction of daily passenger traffic.

During the holiday period, according to the length and nature of different holidays, the daily passenger flow change law varies from holiday to holiday. Four important holidays (cars with less than seven seats travel free on highways) are selected: Spring Festival, Ching Ming Festival, Labor Day, and National Day. The traffic volume of seven days before and seven days after the holiday are also studied. There are total 15 days flow, the abscissa number "1–7" means seven days before the holiday, and the abscissa "8" means the holiday, the abscissa "9–15" means seven days after the holiday, as shown in Figure 2.

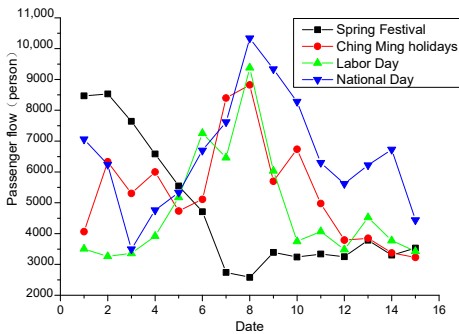

**Figure 2.** Diurnal variation of holiday passenger flow.

There are significant differences in the daily passenger flows variation patterns of the four holidays, see Figure 2. Among them, the passenger flow showed a downward trend in the first seven days before the Spring Festival, then the New Year's Eve and the Spring Festival day were the lowest valleys of passenger flow. After that, the passenger flow rebounded slightly, but it remained at a low level. Less intercity travel may be due to a large number of migrants in this region returned to their hometown in the early Spring Festival. The passenger flow of Ching Ming Festival and Labor Day have similar variation, showing a rising passenger flow before the seven days of holiday, the peak appears on that day of the holiday, and next gradually declines to a stable level after seven days. The Mid-Autumn Festival is on 24 September 2018, and the passenger flow before the National Day holiday first declined slightly then rose straight to the highest point on 1 October, which was affected

by the return journey after the Mid-Autumn Festival. The traffic volume in the next seven days is still greater than that in other holidays because of the longer National Day holiday; there is a downward trend during the seven-day holiday.

### 3.3.2. Hourly Passenger Flow Characteristics

The natural habits of travelers determine the fluctuation of the passenger traffic of the intercity shuttle at different times of the day. In order to explore the hourly changes in passenger flow between different weekdays and normal weekends, the hourly passenger flow of the Shenzhen–Guangzhou intercity shuttle from 10 September 2018 to 16 September 2018 (Monday to Sunday) was selected for research. Meanwhile, the four holidays in 2018 are selected to study the hourly passenger flow of the Shenzhen–Guangzhou intercity shuttle. According to the operation time of the passenger line, the time scale is divided into 16 time periods.

It can be seen from Figure 3 that the hourly variation of passenger flow from Monday to Sunday presents roughly "M" type characteristics, both of which have two peak periods; the morning peak appears at 9:00 to 10:00, the evening peak at 15:00 to 16:00, the passenger flow is relatively flat during 10:00 to 15:00. The hourly flow changes on Tuesday and Thursday are roughly the same, at a relatively low level; the hourly passenger flow curve on Monday and Friday is approximately symmetrical, the passenger traffic on Monday is high in the morning, but is high in the afternoon on Friday. This may be because after the weekend break, more travelers make an intercity trip on Monday morning, and more travelers make intercity trips on Friday afternoons in advance; the hourly passenger flow changes on Saturday and Sunday are similar, and the passenger flow is at a high level every hour all day. This is because there are more travelers conducting intercity trips to visit relatives and friends on weekends.

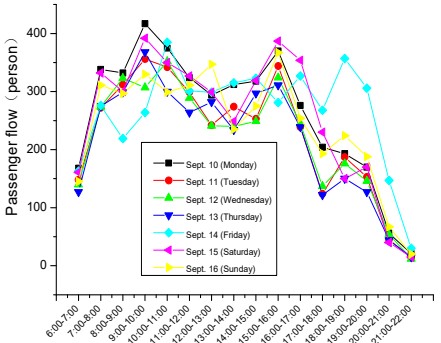

**Figure 3.** Hourly flow of different days of the week.

It can be seen from Figure 4 that the hourly passenger flow change characteristics between different holidays are significant. The hourly flow change trend during the Ching Ming Festival and National Day is about the same, showing the "M" type curve. The peak passenger flow in the morning is greater than in the afternoon. The passenger flow in the morning and evening is small. The hourly curve of the Labor Day is roughly symmetrical as the Ching Ming Festival and the National Day (its passenger flow is small in the morning, and large in the afternoon). It may be because May 1 (Labor Day) is the last day of the holiday, most of the travelers choose to return in the afternoon, while April 5 (Ching Ming Festival) and October 1st (National Day) are both the first day of the holiday, and the traveler is more willing to choose the morning trip; the hourly passenger change curve of the Spring Festival is obviously lower than other three holidays, because it is the first day of the Spring Festival so most people choose to reunite with their families at home instead of carrying out intercity trips. Moreover, most of the residents in Guangzhou and Shenzhen are migrant population, thus they are usually not in the Pearl River Delta during the Spring Festival holiday.

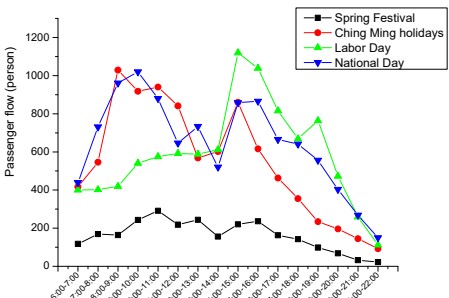

**Figure 4.** Hourly passenger flow of different holidays.

## 4. Non-holiday Passenger Flow Prediction

### 4.1. Data Preparation

The passenger traffic of the Shenzhen–Guangzhou intercity shuttle in 2017 and 2018 are measured on a daily time scale. In order to distinguish between "the day of the week" attributes of different days, the numbers 1, 2, 3, 4, 5, 6, 7 are respectively indicated on Monday, Tuesday, Wednesday, Thursday, Friday, Saturday, and Sunday; holiday attributes, with the number 0 for non-holiday, the numbers 1, 2, 3, 4, 5, 6, 7 represent New Year's Day, Spring Festival, Ching Ming Festival, Labor Day, Dragon Boat Festival, Mid-Autumn Festival, and National Day holiday.

The 57-day passenger flow data from 2017 to 2018 were deleted to form the basic data. The single-sample Kolmogorov–Smirnov test was performed on the basic data using SPSS software. The bilateral progressive significance was less than 0.05, which did not meet the normal distribution. Therefore, the SPSS software is used to find the quartile of the passenger flow data instead of $3\sigma$ criteria, and the upper and lower limits of the normal value of the daily passenger flow data are obtained (1471, 7540), and the 11 daily traffic abnormal value data not belonging to the interval is deleted. The processed sample data is 655. The sample data is normalized by the mapminmax function in MATLAB, so that the value of the sample data is in the range of $(-1, 1)$.

### 4.2. Non-Holiday Passenger Flow Prediction Model

#### 4.2.1. Passenger Flow Prediction Model

The passenger flow prediction model is mainly divided into three categories: Statistical model, nonlinear model, and mixed prediction model. BP neural network is a neural network model that uses error back propagation algorithm. It is also the most widely used neural network, with strong nonlinear mapping and complex logic computing ability [50–52]. The BP neural network structure includes the input layer, the hidden layer, and the output layer. The training process mainly includes signal forward transmission and error back propagation. The signal is forwardly transmitted during the learning process. When the expected output cannot be obtained, the error propagates backward. By modifying the weights of the various neurons, the error is minimized until the desired output is obtained [53–56].

According to the analysis of passenger flow time characteristics above, the daily passenger flow of the intercity shuttle is fluctuating violently, the holiday passenger flow is much larger than the non-holiday passenger flow, and it has typical characteristics such as non-linearity and non-stationarity. BP neural network is a typical nonlinear passenger flow prediction model. Therefore, BP neural network is used as the basis for passenger flow prediction of intercity passenger transport.

Combined with the characteristics of the sample data of this study, a three-layer BP neural network model with a hidden layer was selected. The traffic volume of the prediction day is output by using the traffic volume of the seven days before the prediction, the year, month, day, and "day of the week" attribute of the prediction date. Therefore, the number of input nodes is $n = 11$, and the number of output nodes is $l = 1$ in this study. The value range of the number of hidden layer nodes is calculated

by the empirical formula [5,14], and the number of hidden layer nodes $m = 10$ when the average relative error is selected in the prediction experiment is minimized. A tan-sigmoid function with a faster convergence speed and a wider output range is selected between the input layer and the hidden layer as a transfer function, a purelin function is selected (input and output values to take any value) as a transfer function between the hidden layer and the output layer. The gradient descent traingdx function of momentum back propagation and dynamic adaptive learning rate is selected to meet the operation requirements of faster training and larger capacity data. This study chose a smaller learning rate of 0.1. Select the more commonly used 0.9 as the momentum factor for this model. In this study, the ratio of the training set, the validation set, and the test set were set at 80%, 10%, and 10%, respectively, as most studies do. Set the training accuracy to 0.1 and set the maximum number of trainings to 10,000 to prevent the training time from being too long. Other unspecified parameters settings follow the default values in MATLAB.

### 4.2.2. Improved Genetic Algorithm-Back Propagation Neural Network (IGA-BPNN) Prediction Model

This paper abandons the selection operation of the roulette method in the traditional genetic algorithm, and adopts the selection operation method based on the fitness function to propose an intercity shuttles passenger flow prediction model based on the improved genetic algorithm (IGA) optimized BP neural network (IGA-BPNN), which mainly includes three parts: The first part is to determine the network structure of BP neural network, determine the number of network layers and the number of nodes in each layer; the second part is to use the improved genetic algorithm to optimize the initial weight and threshold of BP neural network, the core of which is the improvement of the selection operation; the third part is to optimize the BP neural network's initial weights and thresholds and then use the optimized network to predict the passenger flow of the intercity shuttle.

Figure 5 shows the IGA optimization process.

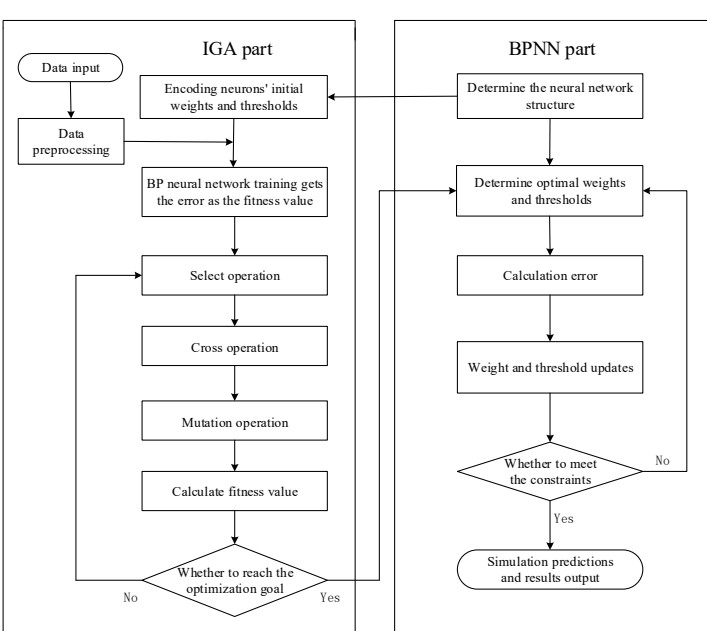

**Figure 5.** IGA-BPNN prediction model flowchart.

The IGA optimization process includes five basic components:

Step1: Chromosome coding. In this paper, the real number coding method is used to encode the input layer, hidden layer, and output layer nodes respectively, which are 11, 13, 1, so there are 156 weights and 14 thresholds. Therefore, when performing real number encoding, the chromosome encoding length is 170.

Step2: Selection of fitness function. The absolute error between the predicted and original value is used as the individual fitness value. If the individual fitness value $i$ is $F_i$, and the corresponding absolute error is $E(X_i)$, the fitness function is as Equation (1), the size of the fitness is a direct manifestation of the individual's performance. The optimization goal of the genetic algorithm is to minimize the individual fitness value until it tends to zero.

$$F_i = E(X_i) \tag{1}$$

Step3: Select operation. The probability that an individual proposed in this paper $p_i$ is selected as Equation (2).

$$p_i = \frac{l/F_i}{\sum\limits_{i=1}^{N} l/F_i} \tag{2}$$

where, $i$ means the individual $i$ ($i = 1, 2, \cdots, N$); $F_i$ means the fitness value of $i$, which used absolute error in this article; $l$ means the adjustment factor.

Step4: Cross operation and mutation operation. Since the code in this paper is a real number code, the real number cross method is selected in the cross operation. According to previous studies, the initial crossover probability can be set to 0.6, and the mutation probability can be set to 0.5.

Step5: Initial population size and maximum evolutionary iteration. The initial population size can be set to 20. In this paper, the maximum evolution iteration is set to a slightly larger value. The fitness curve is observed after the training is completed; the most suitable evolutionary algebra is then selected.

*4.3. Passenger Flow Prediction Result Analysis*

4.3.1. IGA-BPNN Predictive Model Results Analysis

According to the above parameters set, the program corresponding to the IGA-BPNN model is written in MATLAB for prediction. In the improved genetic algorithm, the maximum evolution algebra is set to 400, and the optimal fitness for chromosome evolution is 155. When the evolution exceeds 300 generations, the fitness value has only a slight change. When the evolutionary algebra exceeds 350, the fitness value is almost unchanged. After obtaining the optimal initial weight and threshold, enter to BP neural network. After learning and training, BP neural network predicts the daily passenger flow of non-holiday intercity shuttle.

In order to more intuitively reflect the difference between the predicted and the actual value, the following comparison between the predicted and original value are made as shown in Figure 6.

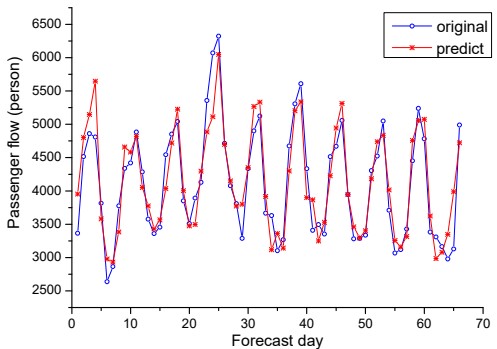

**Figure 6.** Predicted and original non-holiday values.

The prediction results are relatively stable. Moreover, the error of the training set and the verification set data is close to the test set, and the BP neural network has no fitting phenomenon in the prediction. Therefore, the IGA-BPNN model of the non-holiday intercity passenger class passenger traffic has a certain reliability.

4.3.2. Comparative Analysis of Multiple Model Prediction Results

The advantages and disadvantages of the prediction result need to be judged by four evaluation indicators. The model prediction results of IGA-BPNN were compared with the prediction results of GA-BPNN (genetic algorithm optimized BP neural network) and BPNN model selected by roulette method and ARIMA model, respectively. As is shown in Table 6.

**Table 6.** Comparison of different model prediction accuracy.

| Prediction Model | MAE | MAPE(%) | RMSE | MSPE |
|---|---|---|---|---|
| IGA-BPNN | 259.18 | 6.43 | 323.90 | 0.08 |
| GA-BPNN | 376.25 | 9.39 | 425.76 | 0.11 |
| BPNN | 746.10 | 18.62 | 838.06 | 0.61 |
| ARIMA | 510.93 | 11.53 | 688.95 | - |

The MSPE of ARIMA model has not been calculated here due to the former three errors are large than IGA-BPNN obviously, so there is "-"of this error.

The prediction error of the IGA-BPNN model is the smallest (mean absolute percentage error MAPE = 6.43% < 10%), and the values of the other three indicators mean absolute error (MAE), root mean square error (RMSE), and mean square percentage error (MSPE) are relatively small (259.18, 323.90, 0.08). The IGA-BPNN prediction model has strong applicability and reliability for passenger flow prediction of non-holiday intercity shuttle.

*4.4. Non-Holiday Space Passenger Flow Prediction*

The same non-holidays are selected to predict and analyze the intercity passenger traffic between nine cities, and the prediction models are verified from the perspective of panel data. This paper selects 22 August 2018 as the prediction date, and obtains the predicted traffic volume of intercity shuttles among nine cities in the Pearl River Delta on 22 August 2018. The average absolute error between the predicted value and the actual value is calculated separately, and the results are shown in Table 7.

**Table 7.** Non-holiday space passenger flow prediction error (%).

| MAPE | Guangzhou | Shenzhen | Zhuhai | Foshan | Huizhou | Dongguan | Zhongshan | Jiangmen | Zhaoqing | Mean |
|---|---|---|---|---|---|---|---|---|---|---|
| Guangzhou | - | 7.30 | 9.47 | 7.10 | 7.64 | 5.08 | 5.52 | 7.68 | 10.65 | 7.56 |
| Shenzhen | 8.21 | - | 5.86 | 14.64 | 6.63 | 5.64 | 7.28 | 8.06 | 6.32 | 7.83 |
| Zhuhai | 6.76 | 5.08 | - | 7.25 | 7.88 | 2.25 | 7.32 | 4.45 | 9.63 | 6.33 |
| Foshan | 7.90 | 9.05 | 6.79 | - | 4.77 | 2.89 | 4.14 | 6.56 | 5.80 | 5.99 |
| Huizhou | 5.96 | 6.72 | 6.98 | 12.18 | - | 5.96 | 6.00 | 8.06 | 6.61 | 7.31 |
| Dongguan | 5.24 | 6.37 | 6.59 | 12.13 | 13.40 | - | 7.42 | 3.86 | 2.84 | 7.23 |
| Zhongshan | 7.64 | 8.45 | 6.40 | 5.12 | 1.70 | 9.30 | - | 4.93 | 3.54 | 5.89 |
| Jiangmen | 6.43 | 9.75 | 4.21 | 7.53 | 3.23 | 3.01 | 8.85 | - | 8.18 | 6.40 |
| Zhaoqing | 8.30 | 3.70 | 6.65 | 6.97 | 4.17 | 6.66 | 6.41 | 3.73 | - | 5.82 |
| Average | 7.06 | 7.05 | 6.62 | 9.12 | 6.18 | 5.10 | 6.62 | 5.92 | 6.70 | 6.71 |

The origin and destination of the intercity shuttle bus cannot be in the same city, so there is "-"between the same cities.

It can be seen from Table 7 that the predicted average absolute error is between 1.70% and 14.64%. The average absolute error of passenger flow prediction between nine cities on 22 August is 6.71%, with about 0.28% differs by the average absolute error of non-holiday passenger flow prediction for Shenzhen–Guangzhou above, which is an acceptable level of error. It indicates the spatial reproducibility of the non-holiday passenger flow prediction model, and its passenger flow prediction model has strong applicability in the non-holiday passenger flow prediction among nine cities in the Pearl River Delta.

## 5. Holiday Passenger Flow Prediction

### *5.1. Holiday Impact Time Analysis*

#### 5.1.1. The Introduction of Holiday

There are two types of holidays in China's legal holidays, including three-day and seven-day holidays. New Year's Day, Ching Ming Festival, Labor Day, Dragon Boat Festival, and Mid-Autumn Festival are three-day holidays. The Spring Festival and National Day are seven-day holidays.

The demand for intercity travel is huge during the holiday period. Some travelers will take pre-departure and postponed return trips to avoid travel difficulties such as purchasing tickets on holidays and traffic jams. Therefore, this article generalizes the flow peak duration before and after the holidays and the duration of the holiday collectively as the holiday impact time.

The holiday impact time can be analyzed according to the historical passenger flow data of the intercity shuttle, determined by the ratio of the daily passenger flow before and after the holiday to the average passenger flow of the same day of the year, and several days before and after the holiday. When the ratio exceeds the set threshold, the period of time is considered to be the holiday impact time. According to previous research results, the threshold is set to 1.2, and each day of the holiday belongs to the holiday impact time.

#### 5.1.2. The Impact time of Each Holiday

First, the daily passenger flow of each holiday and the days before and after the holiday of a single intercity shuttle line of Shenzhen–Guangzhou are selected to study, then the ratios are calculated as Figure 7. In the three-day holiday, New Year's Day is selected as an example, and the daily passenger flow of the holiday and the 10 days before and after the holiday for a total of 23 days are selected for research.

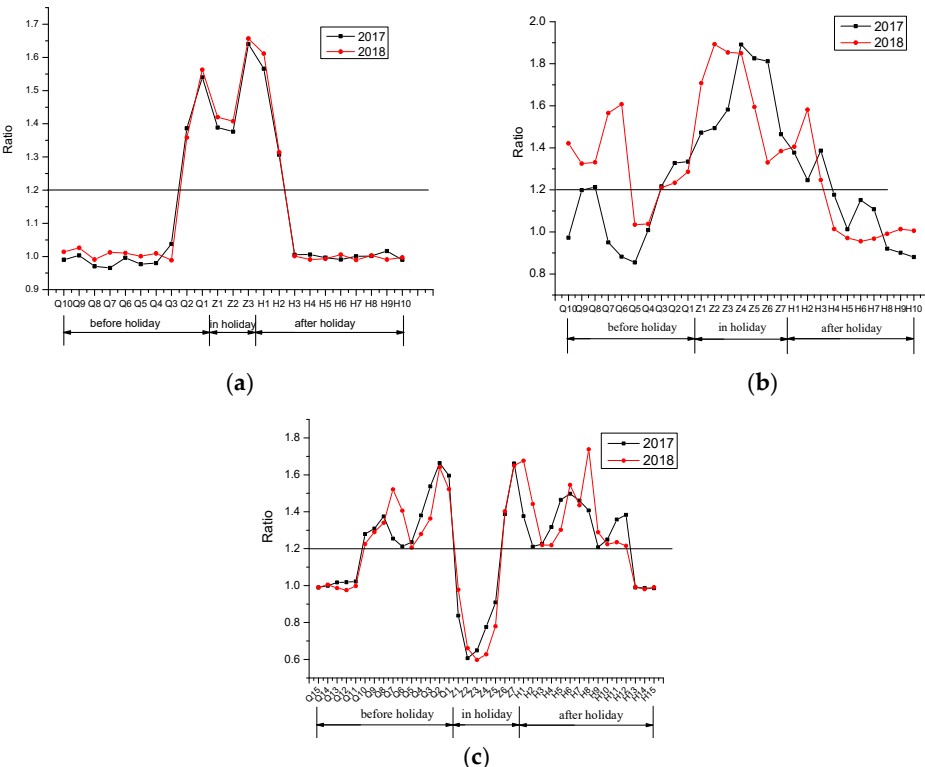

**Figure 7.** Comparison of average passenger flow rate between passenger flow and "the same day of the week" during the year. (**a**) New Year's holiday; (**b**) National Day holiday; (**c**) Chinese New Year holiday.

As shown in Figure 7a, the ratio of New Year's Day holidays in 2017–2018 is greater than 1.2. According to the above-mentioned influence time judgment criteria, two days before the New Year holiday, three days of New Year's Day holiday, and two days after New Year's Day holiday, a total of seven days is the New Year's Day holiday impact time. After similar analysis, the holiday impact time of the Ching Ming Festival, Labor Day Holiday, Dragon Boat Festival Holiday, and Mid-Autumn Festival Holiday are the same as that of the New Year's Day holiday, which is two days before the holiday, three days of the holiday, and two days after the holiday, for a total of seven days. During the three-day holiday, the peak of passenger outflows was concentrated, and the difference in passenger traffic on each day was small.

We select the 10 days before the National Day holiday in 2017–2018, the seven days of the National Day holiday, and the seven days after the National Day holiday to study. The results are shown in Figure 7b, showing the three days before the National Day holiday, seven days of the National Day holiday, and three days after the National Day holiday, a total of 13 days is the National Day holiday impact time. The traffic volume on the two days before and after the holiday was relatively large, and the traffic on other days was relatively average.

A total of 37 days before the Spring Festival holiday in 2017–2018, seven days in the Spring Festival holiday, and 15 days before and after the Spring Festival holiday were selected for the study. The results are shown in Figure 7c. The calculation can be seen 10 days before the Spring Festival holiday, seven days of the Spring Festival holiday, and 12 days after the Spring Festival holiday, a total of 29 days is the Spring Festival holiday impact time. The daily traffic volume before and after the holiday is large, but the passenger traffic during the holiday is lower than the annual average daily passenger flow, and then grows faster.

### 5.2. Holiday Passenger Flow Prediction Model

#### 5.2.1. Prediction Model

China's legal holidays contain, in total, 29 days (about 7.95%), a lower proportion. The number of holiday passenger data is limited. Due to the limitation of the number of holiday passenger flow data, many scholars' prediction accuracy of holiday passenger flow data is still not ideal. And although some scholars have obtained a large number of holiday historical passenger flow data, also using machine learning algorithms to predict future holiday flow, the data of holiday passenger flow in different years are absolutely different, especially the characteristics of holiday passenger flow between years that are far away, so the prediction accuracy cannot be guaranteed.

Therefore, the holiday passenger flow prediction model is combined with the holiday background traffic volume and the holiday passenger flow fluctuation coefficient. It is divided into three parts: The first part is to use the non-holiday passenger flow prediction model to predict the background passenger flow in each day of the holiday impact time; the second part is to combine the historical passenger flow data to determine the holiday passenger flow fluctuation coefficient of each day in the holiday impact time; in the third part, the holiday background passenger traffic is multiplied by the holiday passenger flow fluctuation coefficient to obtain the predicted value of the holiday passenger flow.

The predictive model can be shown as:

$$Y_{ij} = \phi_{ij} y_{ij} \tag{3}$$

where, $i$ represents the seven holidays mentioned above respectively ($i = 1, 2, \cdots, 7$); $j$ means different dates during the holiday ($j = -10, -9, \cdots, 0, 1, \cdots, 18$); $Y_{ij}$ means the predicted value of daily passenger flow within the duration of the holiday; $\phi_{ij}$ means the fluctuation coefficient of holiday passenger flow on each day of the holiday; $y_{ij}$ means the background traffic of each day within the holiday.

### 5.2.2. The Fluctuation Coefficient

The fluctuation coefficient of passenger flow for each day of different holidays is directly related to the fluctuation coefficient of historical passenger flow, and it is also related to the coefficient of variation of the average daily passenger flow in the forecast year.

Therefore, the fluctuation coefficient of the holiday passenger flow is composed of two parts: The first part is the ratio of the passenger flow for each day of holiday duration in the previous year to the average passenger flow of the "day of the week" to which it belongs, and the second part is the variation coefficient of the average holiday passenger flow between two years. The calculating formula is as follows:

$$\phi_{ij} = \varphi_{ij}k_i \tag{4}$$

where, $\varphi_{ij}$ means the ratio of the passenger flow for each day of holiday duration in the previous year to the average passenger flow of the "day of the week" to which it belongs; $k_i$ means the variation coefficient of the average holiday passenger flow between the two years.

### 5.3. Prediction Results and Model Verification

#### 5.3.1. Prediction Result

Table 8 and Figure 8 show the results. The background passenger traffic of each holiday is predicted according to the IGA-BPNN intercity shuttle non-holiday passenger flow prediction model above and predicts the passenger flow of each day in 2018. Generally, the average of the holiday passenger flow prediction MAE, MAPE, RMSE, and MSPE is at an ideal level. There are differences in the average errors of the predicted values within the duration of different holidays, but the differences are not very significant. Therefore, the prediction accuracy of the holiday prediction model is relatively high, and it has superior prediction performance in the case of less passenger traffic data.

**Table 8.** Holiday passenger flow prediction error.

| Holiday | MAE | MAPE(%) | RMSE | MSPE |
|---|---|---|---|---|
| New Year's holiday | 312.77 | 4.50 | 319.60 | 0.05 |
| Spring Festival | 373.85 | 6.54 | 422.29 | 0.07 |
| Ching Ming holidays | 514.44 | 7.36 | 539.38 | 0.08 |
| Labor Day Holiday | 490.83 | 7.24 | 528.66 | 0.08 |
| Dragon Boat Festival | 488.22 | 6.96 | 538.35 | 0.07 |
| Mid-Autumn Festival | 382.92 | 5.34 | 440.78 | 0.06 |
| National Day holiday | 391.75 | 5.23 | 484.78 | 0.06 |
| Average | 405.96 | 6.20 | 460.42 | 0.07 |

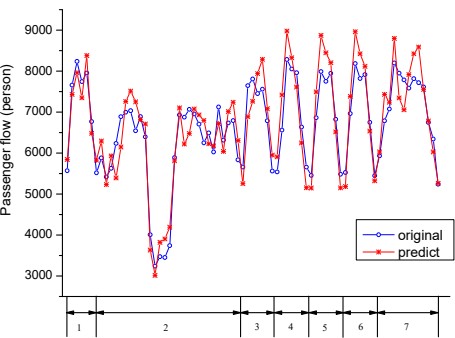

**Figure 8.** Predicted and original holiday values. 1 means New Year's holiday, 2 means Spring Festival, 3 means Ching Ming holidays, 4 means Labor Day Holiday, 5 means Dragon Boat Festival Holiday, 6 means Mid-Autumn Festival, 7 means National Day holiday.

5.3.2. Predictive Model Verification

Similarly, the model is validated with the support of the holiday passenger panel data with different routes.

The passenger traffic prediction value of May 1, 2018 is selected for verification, and the average value of various prediction errors is calculated (Table 9). The mean absolute error (MAE) and root mean square error (RMSE) are relatively higher than the single-line above, the decrease may be due to the smaller passenger flow on the other intercity shuttles than that of Shenzhen–Guangzhou route; while the mean absolute percentage error (MAPE) and the mean square percentage error (MSPE) are not much different from that of Shenzhen–Guangzhou line, which indicates that the prediction result is stable and the prediction accuracy does not fluctuate much, which is at a relatively ideal level, proving the universality and replicability of this model. It can be seen that the prediction model proposed in this paper can overcome the limitation of the lack of historical holiday data and can be applied to the passenger flow prediction of the different intercity shuttle routes in megalopolis on holidays.

**Table 9.** Average holiday passenger flow prediction error based on panel data.

| Holidays | MAE | MAPE(%) | RMSE | MSPE |
|---|---|---|---|---|
| New Year's holiday | 314.21 | 5.44 | 204.33 | 0.06 |
| Spring Festival | 265.36 | 7.52 | 311.52 | 0.09 |
| Ching Ming holidays | 357.16 | 6.16 | 299.37 | 0.06 |
| Labor Day Holiday | 291.34 | 4.11 | 258.34 | 0.06 |
| Dragon Boat Festival | 322.48 | 6.31 | 325.71 | 0.08 |
| Mid-Autumn Festival | 304.01 | 4.92 | 280.90 | 0.07 |
| National Day holiday | 297.88 | 5.77 | 244.64 | 0.07 |
| Average | 307.49 | 6.63 | 340.75 | 0.07 |

*5.4. Application of Holiday Passenger Flow Prediction Analysis*

Different city pairs have different rules of intercity shuttle due to the difference in the permanent resident number and the economic development level. According to the predicted passenger flow, operating companies can set up transportation plans and resource allocation reasonably to balance the corporate profits and holiday travel needs, and the departure schedule of the intercity shuttle can be appropriately optimized to adjust to maximize the operational efficiency and the passenger satisfaction. Operating enterprises can grasp the changing rules of the passenger flow in the future to reserve the vehicles and personnel in advance according to the specific flow growth. Passenger stations can achieve emergency dispatch measures of vehicles on different intercity shuttles and respond to the surge in passenger flow on a certain line according to the accurate prediction of passenger flow peaks during holidays, improving the efficiency of passenger flow evacuation in the station. Travelers can also adjust their travel plans according to their actual optimization, and obtain higher travel service quality by purchasing tickets in advance, adjusting travel time, and shifting shuttle bus to stagger the peak period.

Specifically, the characteristics of passenger travel on holidays are quite different from those on non-holidays. Operating enterprise can realize the organization and management of shuttle routes from the aspects of popular route capacity guarantee and time-differentiated capacity reserve. From the spatial level, the capacity organization is organized by different lines; from the time level, the allocation of capacity resources is carried out by different holidays and time periods.

(1)    Popular route capacity optimization:

There are huge differences in passenger flow of different intercity shuttle corridors. From a regional perspective, the large passenger flow in various cities is mainly concentrated between intercity shuttle lines such as Guangzhou, Shenzhen, Foshan, and Dongguan. These four cities have a large number of travel demands, of which the passenger flow between Guangzhou and Shenzhen is the most, and the passenger flow from Shenzhen to Guangzhou is slightly larger than the passenger flow from

Guangzhou to Shenzhen. It is necessary to increase the number of passenger departures to ensure that the corridors between popular cities do not surpass capacity.

(2)    Time differentiated capacity reserve:

The characteristics of spatial passenger travel in different holidays are quite different, so that differentiated organization arrangements should be implemented according to different holidays. During the Spring Festival, a large number of migrant workers will return to their hometowns collectively. The hot travel routes before the holiday are mainly spread to the surrounding areas centered on megacities, and the hot return routes after the holiday mainly gather from small and medium cities to central cities. Tomb-sweeping trips are mostly for tourists from urban areas returning to cemeteries to sweep graves. Temporary routes from the urban area to surrounding cemeteries can be opened to meet residents' needs for grave sweeping. During Labor Day and National Day, the purpose of travel is mainly for leisure travel and visiting relatives. Special tourist shuttles connecting major transportation hubs to popular tourist attractions can be added to meet the transportation needs of tourists. Passenger transport routes should be adjusted dynamically according to the difference in passenger transport demand.

(3)    Differentiated capacity equipment at different times during holidays:

Even on the same holiday, the traffic characteristics at different times are also very different. Passenger flow peaks often occur before holidays, and there is a problem of insufficient capacity before holidays. Therefore, pre-holiday capacity allocation needs to be paid attention to. Capacity needs to be reserved before the peak of holiday travel to ensure the overall travel capacity demand during peak travel on holidays. Especially during the Spring Festival holiday, key cities such as Guangzhou and Shenzhen will have the phenomenon of export peaks in the early period of the holidays, empty cities in the middle of the holidays, and peak return trips in the latter part of the holidays. Therefore, the transportation resources of the holidays should be developed in advance, and the remaining transportation capacity in the middle of the holidays should be rationally allocated to the early and late holidays.

## 6. Conclusions

This paper analyzed the spatiotemporal characteristics of intercity shuttles passenger flow in the Pearl River Delta and established separate passenger flow prediction models on non-holiday and holiday, which were then validated based on panel data. Firstly, the spatial and temporal characteristics of passenger flow in the intercity shuttle were analyzed in detail. Then the passenger flow prediction model of intercity shuttle based on BP neural network was proposed. Next a passenger flow prediction model on holidays combining holiday background traffic flow and holiday flow fluctuation coefficient was proposed based on the non-holiday passenger flow prediction model. Finally, the two prediction models were verified by historical passenger flow data. The specific results and conclusions of this paper are as follows:

First, the spatial and temporal characteristics of passenger flow in the intercity shuttle were analyzed. The passenger line flows in different cities have a direction imbalance. The main passenger flow direction is concentrated between cities with a large number of permanent residents and a good level of economic development, and there are also differences in passenger flow between the opposite directions of passenger transport. The same "day of the week" traffic between different weeks is closely relative, and the correlation is relatively large; the passenger flow varies greatly between holidays, and the nature of this difference is the different travel demand between different holidays. The passenger traffic on the weekend is basically higher than the weekday; the hourly passenger flow on the holiday is much larger than that on the weekend and weekday, having obvious holiday characteristics; there are also obvious differences in the passenger flow between different holidays.

Second, a passenger flow prediction model for intercity shuttle on non-holiday was established. Based on the analysis of the space-time characteristics of the passenger flow in the intercity shuttle, aiming at the shortcomings of the existing traditional BP neural network prediction model, the genetic algorithm is used to optimize the initial weight and threshold of the BP neural network, and the selection operation of genetic algorithm is improved based on fitness, and the passenger flow prediction model (IGA-BPNN) of the intercity shuttle based on BP neural network with improved genetic algorithm is finally formed. The model is verified by historical passenger flow data. The results show that the prediction accuracy (6.43%) of this model is higher than that of pure BP neural network prediction model (BPNN 9.39%) and the traditional genetic algorithm optimization BP neural network prediction model (GA-BPNN 18.62%).

Third, a passenger flow prediction model for intercity shuttle on holiday was established. Aiming at the shortcomings of the lack of historical holiday passenger flow data of the intercity shuttle, based on the passenger flow prediction model on non-holiday, a passenger flow prediction model on holiday combining holiday background traffic flow and holiday passenger flow fluctuation coefficient is proposed. This model is verified using panel data of the intercity shuttle among cities in the Pearl River Delta. The results show that the holiday passenger flow prediction model has higher prediction accuracy and strong applicability when there is less passenger traffic data in the holiday. Thus, this model can overcome the limitation of the lack of historical holiday data and can be applied to the passenger flow prediction of the different intercity shuttle routes in the megalopolis.

If the method could be replicated for other megalopolises in different countries with different cultural backgrounds, the study has certain limitations:

First of all, the research scope, Pearl River Delta, has a large number of factories and manufacturing industries. During short holidays, intercity traffic is frequent, which is the main tourist attraction. During the long holidays, especially the Spring Festival, most people choose to leave and go back to remote hometowns due to the influence of traditional concepts.

Next, there are few holiday data, only including the holiday data of more than 50 days in two years. The forecast results will be more accurate if there are more holiday data.

Furthermore, the algorithm time is relatively long, and the adaptive adjustment problem of BP network parameters also needs further study.

Due to the limitations of the data sources and the theoretical level, this paper has done some analysis on the passenger flow prediction of the intercity shuttles considering only some of the influencing factors (the historical passenger flow data and the time attribute of the prediction date). It can also be applied to predictions of other large cities in other megalopolises. However, many other influencing factors (extreme weather and large events) have not been considered, thus the method may not be applied on these conditions. These factors can be carried out in the future to enhance the applicability and accuracy of the prediction model. Besides, the less iterative efficiency of the genetic algorithm in the iterative process needs to be improved in some predictions with strong real-time requirements. As the data is enriched and refined in the future, the prediction accuracy of the model will increase.

**Author Contributions:** Conceptualization, B.X.; methodology, B.X.; writing—original draft preparation, Y.S.; writing—review and editing, Y.S.; visualization, Y.S.; data curation, X.H.; visualization, X.H.; investigation, L.Y.; validation, L.Y.; supervision, G.X.; software, G.X. All authors have read and agreed to the published version of the manuscript.

**Funding:** This research was funded by the National Natural Science Foundation of China, grant number 71974043, 71473060 and 91846301.

**Acknowledgments:** The authors would like to thank the anonymous reviewers for their valuable comments.

**Conflicts of Interest:** The authors declare no conflict of interest.

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
