# Peer review of "Travel Characteristics Analysis and Passenger Flow Prediction of Intercity Shuttles in the Pearl River Delta on Holidays"

_sustainability, doi:10.3390/su12187249_

Round 1
Reviewer 1 Report
23 references are not sufficient for the literature review. Authors should use more sources covering a border range of studies on the topic considering also studies coming from other countries. So, I suggest authors use at least 40 references for their paper.
- Ghalehkhondabi, I., Ardjmand, E., Young, W.A. and Weckman, G.R. (2019), "A review of demand forecasting models and methodological developments within tourism and passenger transportation industry", Journal of Tourism Futures, Vol. 5 No. 1, pp. 75-93. https://doi.org/10.1108/JTF-10-2018-0061
- Investigating Effect of Holidays on Daily Traffic Counts: Time Series Approach https://journals.sagepub.com/doi/abs/10.3141/2019-04
- https://doi.org/10.1080/03081060.2012.673272
The paper needs proof reading. Sentence are quite often very long and not clear. See sections 3.2.1, 3.2.2., 5.1.2 (e.g. Count the average daily departures of intercity shuttle in the Pearl River Delta in 2017-2018)
In section 3.2.1, authors are referring to the data that has been not mentioned in the body text. I would suggest to include a summary table of spatial shift.
In section 3.3.1, authors discussed about variation of passenger flow in the Shenzhen-Guangzhou passenger transport line comparing with holiday passenger which needs more clarification. For instance, it should be explained whether the passenger flow in Fig1. is an aggregated flow of different transport mode (e.g. car, train) or not?
In Table 2,
- ID column can be eliminated since it does not bring additional information to readers.
- Outbound time can be change to the day of the week and time of day (TOD)
It’s not clear that the Fig7 is referring to which passenger flow corridor. If it’s an aggregated figure, then it should be split into 9 separate figures and results should be further discussed. Furthermore, figures presented in section 5.1.2 are not clear to which passenger flow is referring. (car passenger flow or intercity shuttle flow)
There is a major inconstancy in modelling no-holiday and holiday passenger flow predictions and the results interpretation. Authors focus on Non-holiday Passenger Flow Prediction Model was on only the Shenzhen-Guangzhou corridor while in Holiday Passenger Flow Prediction Model it seems that in Holiday Passenger Flow Prediction Model focus was on regional or national level.
Section 5.4 "application of holiday passenger flow prediction analysis" is very generic and there is no specific highlight for decision-making process about travel demand management for stakeholders such as operating enterprise in Pearl River Delta and in particular for those 9 studied corridors. This section needs a major revision based on analysis results.
Reviewer 2 Report
This article addresses an interesting and up-to-day issue of optimizing the operation of transport companies by better predicting passenger flows on different times (weekdays and holidays) on the example of intercity shuttles in the Pearl River Delta. related to safety concerns about urban mobility in the past-pandemic cities. This topic is tackled by building upon the theoretical background of predicting traffic flows with the use of different methods including statistical models, machine learning and others. Despite the fact that there are some unclarities about the aim of the paper (discussed later), the proposed method of predicting the passenger flows seems to add value to this field of research and as authors point to, will only grow more accurate with the use of longer-term datasets (if available to obtain).
The used data set and methodology are thoroughly explained and discussed. Although the data set is not provided and thus it would be hard to replicate the presented results – especially as only examples are provided to prove the validity of undertaken approach.
The limitations of study are discussed but only briefly. If the method could be replicated for other megalopolises in different countries with different cultural background (e.g. different number o holiday days in different cultural circles) they should be explained in bigger detail.
Some smaller things also may require a bit of reworking, like the fact that the goal of the paper is not clearly stated – neither in abstract or in the Introduction. The intention of the authors is well described with the reasons for undertaking that particular topic in the section 1. Introduction but it feels more like a summary of the second section 2. Literature review with adding short description about the used methods. What more there are no references in the introduction, even though the authors are clearly making some references like in line 36 or lines 77-78.
The structure of the paper is not described in the introduction.
Lines 93-94 are nearly literally stating the same as lines 77-78.
Section 2 Literature review ends with what could the closest description of the goal so far (lines 174-176) – is it possible to have such clear cut description of what the authors want to achieve closer to the beginning of the paper?
The literature review provide a good explanation for the why (choice of topic) and how (choice of method) authors decided to deal with the issue of intercity shuttle traffic flows predictions. Perhaps reworking or combining section 1 with the section 2 could improve the quality of the paper and at the same time allow to remove some of the repetitions from the text?
The references are relevant and up-to-date.
The language used is mostly correct and allows the reader to follow the authors’ argument, but at the same time there are some minor spelling or editorial mistakes (like line 14, 21, 40-42, 45, 48-50, 53, 59, 72, 75, 102, 106, 127, 141-142, 160,164, 165, 171, 172, 185, 210, 213, 217, 219, 229, 237, 242, 246, 272,281-282, 290, 297-298, 391, 310, and so on) some of may be removed in the process of text editing and proofreading but some will require the attention of the authors – a spelling check would be recommended.
Reviewer 3 Report
The paper has limited interest. The conclusions must be improved. The introduction is not clear. Moderate English changes are required .

Round 2
Reviewer 3 Report
The authors have made the changes needed.
Author Response
We thank the reviewer for providing comments and suggestions for revising the paper, and we are glad to know that the reviewer recognized our efforts. The comments/suggestions are very helpful in improving the quality and clarity of our manuscript. Again, we sincerely appreciate the reviewer for the helpful guidance.